# Cluster and survival analysis of UK biobank data reveals associations between physical multimorbidity clusters and subsequent depression

Lauren Nicole DeLong [1] ✉, Kelly Fleetwood [2], Regina Prigge [2], Paola Galdi [1], Bruce Guthrie [3] & Jacques D. Fleuriot [1,3]

## Abstract

**Background** Multimorbidity, the co-occurrence of two or more conditions within an individual, is a growing challenge for health and care delivery as well as for research. Combinations of physical and mental health conditions are highlighted as particularly important. Here, we investigated associations between physical multimorbidity and subsequent depression.
**Methods** We performed a clustering analysis upon physical morbidity data for UK Biobank participants aged 37–73. Of 502,353 participants, 142,005 had linked general practice data with at least one baseline physical condition. Following stratification by sex (77,785 women; 64,220 men), we used four clustering methods and selected the best-performing based on clustering metrics. We used Fisher's Exact test to determine significant over-/under-representation of conditions within each cluster. Amongst people with no prior depression, we used survival analysis to estimate associations between cluster-membership and time to subsequent depression diagnosis.
**Results** Our results show that the $k$-modes models perform best, and the over-/under-represented conditions in the resultant clusters reflect known associations. For example, clusters containing an overrepresentation of cardiometabolic conditions are amongst the largest (15.5% of whole cohort, 19.7% of women, 24.2% of men). Cluster associations with depression vary from hazard ratio 1.29 (95% confidence interval 0.85–1.98) to 2.67 (2.24–3.17), but almost all clusters show a higher association with depression than those without physical conditions.
**Conclusions** We show that certain groups of physical multimorbidity may be associated with a higher risk of subsequent depression. However, our findings invite further investigation into other factors, such as social considerations, which may link physical multimorbidity with depression.

## Plain language summary

Multimorbidity occurs when an individual has two or more diseases. Previous multimorbidity research often focused on physical disease, but the links between physical and mental health are increasingly recognised as important. Here, we investigate whether certain patterns of physical conditions affect the risk of developing depression. To do so, we use computational methods to identify groups of physical diseases that co-occur. We identify several groups and then explore whether participants in each group were diagnosed with depression. Although risk of depression varies across groups, we show that those with physical diseases have a higher risk than those without. We recommend further studies to investigate other factors which may link physical disease to depression.

Multimorbidity, the simultaneous occurrence of two or more long-term conditions in an individual is increasingly common as populations age, and it challenges existing health systems[1–3]. Multimorbidity is more common with increasing age, in women and in the less affluent[4,5]. Studying the co-occurrence of multiple long-term conditions in the same individual has the

potential to inform understanding of disease causation and support planning of current and future health and care services[6–8].

Depression affects an estimated 6% of people worldwide and is ranked by the World Health Organisation as one of the most burdensome diseases[9–11]. There is strong evidence that depression co-occurs with other

[1]Artificial Intelligence and its Applications Institute, School of Informatics, University of Edinburgh, Edinburgh, UK. [2]Usher Institute, University of Edinburgh, Edinburgh, UK. [3]Advanced Care Research Centre, Usher Institute, University of Edinburgh, Edinburgh, UK. ✉e-mail: L.N.DELONG@sms.ed.ac.uk

mental health disorders[12,13], and several ongoing studies aim to identify potential shared mechanisms[12,14]. However, previous studies have also found depression to be more common in people with particular chronic physical illnesses, such as cardiovascular disease (CVD)[15], multiple sclerosis[16] and inflammatory bowel disease[17]. Physical ill-health might cause depression because it creates psychological disturbance through 'biographical disruption' that threatens a sense of identity, or because of impact on physical or social function. Alternatively, physical conditions may cause depression through intermediate biological processes, like inflammation[16,18], in which case we might expect that different combinations or patterns of physical conditions would be more strongly associated with depression than others.

Several studies have used cluster analyses to identify common patterns of physical conditions[19–22], typically using one method, such as agglomerative hierarchical clustering (AHC)[19,20,23], $k$-medoids[21,24], latent class analysis (LCA)[22,25–27] or $k$-means approaches[28–31]. Additionally, since morbidity data are binary (a person has a condition or does not), some common clustering methods are inappropriate since they use similarity measures incompatible with categorical data[19,32]. Therefore, in this study, we aim to explore and compare the use of four independent clustering methods appropriate for binary data and to examine whether certain groups of physical conditions are associated with the subsequent diagnosis of depression.

We find that the $k$-modes[33,34] clustering method consistently performs best according to several clustering metrics. Additionally, identified clusters align with known associations, such as cardiometabolic conditions. We also find varying levels of association with subsequent depression. Notably, almost all clusters show a higher association with depression than those without physical conditions. We conclude that further investigation is needed into other factors that link physical disease to depression.

## Methods
### Data selection and pre-processing
We used data from UK Biobank (UKB)[35]. Participants aged 37–73 years attended a baseline assessment during 2006–2010, which collected data on demography, lifestyle habits, health conditions and a range of physical and laboratory measurements. Participants provided written informed consent at baseline for both the inclusion of their data within UKB and for the linkage of their data to national datasets including general practice (GP) (primary care), hospital, cancer registry and death records[35]. We conducted our study based on the UKB's generic ethical approval as a research tissue bank from the NHS North West Research Ethics Committee (21/NW/0157). This type of approval covers our research purposes, so no further ethical approval was necessary. All bona fide researchers in academic, commercial and charitable settings can apply to use the UKB resource for health-related research in the public interest (www.ukbiobank.ac.uk/register-apply/). The study was conducted using the UKB resource under application number 57213, and all researchers who accessed the data are listed as collaborators on the application.

To robustly ascertain a broad range of long-term conditions, our study population included participants with a continuous GP record from at least a year before to at least one day beyond their baseline assessment. We excluded records from the UKB extract of the vision practice management system in England because the extraction process excluded participants who died prior to data extraction. We also excluded participants who withdrew from the study (Supplementary Fig. 1).

We ascertained the presence of depression and 69 long-term physical health conditions at baseline using data from the baseline visit and from linked GP, hospital, and cancer registry records based on previously published lists[36,37] (Supplementary Data 1). The UK National Health Service limits registration to one practice at a time, and GP records transfer between practices so should capture an individual's entire medical history. However, available hospital and cancer registry records began at different times for England, Wales and Scotland. Therefore, we used all GP records up to baseline assessment date, and to be consistent across countries, we used hospital and cancer registry records within eight years before baseline assessment date. We used published codelists to identify diagnoses from GP

records using Read V2 and CTV3 diagnosis codes, hospital records using ICD-10 diagnosis codes and OPCS-4 procedure codes, and cancer registry records using ICD-10 codes[37]. We similarly ascertained depression during follow-up using information from GP, hospital and death records. Eligible participants with no history of depression prior to baseline were followed up to the earliest of depression diagnosis, death or the end of their available GP or hospital records.

### Models and metrics
We explored the suitability of four methods ($k$-modes[33,34], $k$-medoids[24], LCA[25] and AHC[23]) to cluster all participants based on binary features denoting the absence/presence of the 69 baseline physical conditions. Participants with no physical conditions at baseline were excluded from the clustering analysis. We additionally clustered separately for men (all 69 conditions) and women (67 conditions since erectile dysfunction and hyperplasia of the prostate are only found in men), as well as in the whole population, because of known sex differences in patterns of individual morbidities and multimorbidity[38,39].

All four clustering methods are designed to operate without predefined classifications, making them ideal for discovering underlying groups within the data[19]. To select cluster number, we used various heuristics, including the elbow method on a scree plot[40] for both $k$-modes and $k$-medoids, and the minimal Bayesian information criterion[41] for LCA. To assess suitability and performance of these clustering methods, we used three performance metrics (Calinski and Harabasz score[42], Davies–Bouldin score[43], and Silhouette score[44] (Supplementary Note 1)), which are appropriate and commonly used for assessing cluster quality in an unsupervised setting. Finally, $k$-modes and LCA are sensitive to differences in initialisation[25,45], so we repeated them five times and compared with other models using the mean and standard deviation across the five experiments.

### Statistics and reproducibility
We used two metrics to analyse over- and under-representation of conditions in each cluster. We designed one metric, the adjusted relative frequency (ARF) to measure the magnitude of over- or under-represented conditions within a cluster, relative to prevalence in the whole cohort. By adjusting for cohort prevalence, the ARF accounts for conditions which are *generally* rare or common, facilitating more effective comparisons between clusters than prevalence alone (Supplementary Fig. 2). For each condition, ARF is calculated as:

$$\text{adjusted relative frequency (ARF)} = \frac{\%\text{with condition in cluster}}{\%\text{with condition in the cohort}}$$

An ARF of exactly one indicates that the condition occurs at the same relative frequency as it does in the entire cohort, and values greater or less than one indicate over- and under-representation, respectively. We used Fisher's Exact Test (two-sided, $\alpha = 0.05$) to evaluate whether over- or under-representation of a condition in each cluster was statistically significant, relative to the respective cohorts ($n = 142,005$ participants in the whole cohort, 77,785 participants in the women-only cohort 64,220 participants in the men-only cohort). We used Bonferroni corrections to account for multiple testing[46–48]. ARF values and adjusted $p$-values are available in Supplementary Data 2. Finally, we visualised and compared statistically significant results on a *bubble heatmap* (Supplementary Note 2). To allow others to conduct similar cluster analyses and visualisations, we made the code available as a software package (https://github.com/laurendelong21/clusterMed)[49].

### Survival analysis to predict depression diagnosis
Using participants without a record of depression at baseline, we applied Cox regression models[50] to evaluate time to depression diagnosis by condition cluster, accounting for death as a competing risk[51]. Participants with no physical conditions at baseline were included as the reference group. We

ran separate models for the whole cohort and for men and women separately, examining associations between cluster membership and subsequent depression. All models were adjusted for baseline age, ethnicity, country of residence and deprivation. The model for the whole cohort was additionally adjusted for sex. Baseline age was included in the models as a continuous variable; all other variables were categorical. Ethnicity was self-reported at baseline, and we categorised it into five groups (Black, Mixed, South Asian, White, and any other ethnic group[52]). Country of residence (England, Wales or Scotland) and area-based deprivation, measured by the Townsend Deprivation Index[53], were derived from participants' home addresses at baseline. We divided the Townsend Deprivation Index into deciles within the entire UKB cohort. A small number of participants (368 women and 435 men) who were missing data on ethnicity, country or deprivation were excluded from the survival analyses.

### Reporting summary

Further information on research design is available in the Nature Portfolio Reporting Summary linked to this article.

## Results

### Performance metrics across various clustering methods

There were 140,956 participants (73,036 women and 67,920 men) with at least one physical condition at baseline who were included in the clustering analysis (Supplementary fig. 1). Performance metrics for each of the four methods explored are reported in Table 1.

Models based on AHC consistently achieved the poorest Calinski and Harabasz, and Davies–Bouldin scores in all three cohorts. Models based on LCA had better metrics than AHC-based models, with particularly high Calinski and Harabasz scores, but the Davies–Bouldin scores were consistently worse in comparison to $k$-modes or $k$-medoids based models. In contrast, the best Davies–Bouldin scores were achieved by the $k$-modes and $k$-medoids based models. However, since the Davies–Bouldin score assesses similarity between the most similar clusters, scores may be optimistic when several clusters only contain a single (or very few) participant(s). This was the case for the $k$-medoids models for the whole and men-only cohorts. The presence of singleton clusters is also concurrent with a larger number of total clusters. Specifically, all three models based on $k$-modes discovered eight clusters, all models using AHC discovered ten clusters, and all models using LCA discovered five or six clusters. In contrast, the $k$-medoids models discovered 25 clusters for the whole population (17 only had one participant), six for women-only (no singletons), and 13 for men-only (seven singletons) (Table 1). Therefore, while $k$-medoids models had comparable

Davies-Bouldin scores to $k$-modes models, the results were less informative and consistent. For each cohort, we therefore selected the best performing $k$-modes model with the highest Calinski and Harabasz score among the five independent runs.

### Differential representation of physical conditions within $k$-modes clusters

Many of the significantly over-represented conditions within several clusters aligned with body systems (Fig. 1) and we therefore used clinical judgement to name the clusters according to the systems or conditions which were most prominent (Supplementary Tables 1–3). The four largest clusters in whole cohort are "*Mixed including cancer*" (27.9% of participants in the cohort), "*Healthy + Rhinitis*" (22.2%), "*CVD + diabetes*" (15.5%), and "*Very extensive morbidity*" (12.5%). For women, the four largest clusters are "*Mixed including cancer*" (29.3%), "*CVD + diabetes*" (19.7%), "*Musculoskeletal (MSK)*" (16.4%), and "*Healthy + Rhinitis*" (15.9%). Finally, for men, the four largest clusters are "*CVD + diabetes*" (24.2%), "*Mixed including cancer*" (20.8%), "*MSK + others*" (19.1%), and "*Healthy + Rhinitis*" (17.2%) (Fig. 2, Table 2).

Of the remaining clusters, there were some similarities across all three cohorts (*e.g.* "*Respiratory*" clusters). Generally, clusters with more participants also tended to have fewer conditions per participant (Fig. 2). For example, the "*Mixed including cancer*" clusters had the lowest mean conditions per participant (whole: 1.77; women: 1.75; men: 1.62). Such clusters may serve as miscellaneous categories for participants with condition profiles that are not easily grouped and/or people with one dominant condition. However, there were also differences. For example, there were clusters which only appeared in the whole population ("*Migraine*") and clusters which only appeared in the sex-stratified cohorts ("*Digestive*" and "*MSK*" clusters).

### Subsequent incident depression per identified cluster

Analysis of time to incident depression diagnosis included 141,001 participants (73,036 women and 67,920 men), excluding 30,770 participants with a history of depression at baseline (20,592 women and 10,178 men). In addition to participants included in the clustering analysis, this analysis also included 30,551 participants with no physical conditions at baseline (16,238 women and 14,313 men) (Supplementary Fig. 1). During an average follow-up of 6.8 years, 5904 (4.2%) participants, including 3574 (4.9%) women and 2330 (3.4%) men, had a new depression diagnosis. Generally, participants with physical conditions at baseline had a higher rate of subsequent depression than participants with no physical conditions at baseline (Table 2, Supplementary Fig. 3).

**Table 1 | Performance metrics, number of clusters, and cluster sizes across four clustering methods**

| Cohort | Clustering method | Calinski and Harabasz score (higher is better) | Davies–Bouldin score (lower is better) | Silhouette score (higher is better) | No. of clusters | No. of singleton clusters | Median participants/ cluster | (Min–max.) participants per cluster |
|---|---|---|---|---|---|---|---|---|
| Whole | LCA | 5567.79 ± 175.06 | 3.80 ± 0.34 | 0.08 ± 0.03 | 6 | 0 | 17,203.5 | (6816–57,564) |
| | AHC | 628.39 | 4.38 | 0.09 | 10 | 0 | 2495.5 | (11–110,366) |
| | $k$-medoids | 1270.00 | 2.48 | 0.08 | 25 | 17 | 1 | (1–55,553) |
| | $k$-modes | 6079.53 ± 1228.07 | 2.43 ± 0.16 | 0.17 ± 0.02 | 8 | 0 | 14,954.5 | (974–39,601) |
| Women-only | LCA | 4098.56 ± 5.23 | 3.53 ± 0.00 | 0.15 ± 0.00 | 5 | 0 | 15,794 | (2373–30,529) |
| | AHC | 524.62 | 4.82 | 0.04 | 10 | 0 | 1374.5 | (2–47,728) |
| | $k$-medoids | 3672.43 | 2.60 | 0.14 | 6 | 0 | 9472 | (3942–28,757) |
| | $k$-modes | 3456.82 ± 555.59 | 2.36 ± 0.10 | 0.16 ± 0.04 | 8 | 0 | 9233.5 | (562–22,787) |
| Men-only | LCA | 3134.68 ± 91.36 | 3.52 ± 0.02 | 0.10 ± 0.00 | 5 | 0 | 13,868 | (6827–16,448) |
| | AHC | 274.95 | 4.18 | 0.20 | 10 | 0 | 1709 | (7–27,353) |
| | $k$-medoids | 1225.80 | 2.17 | 0.11 | 13 | 7 | 1 | (1–25,595) |
| | $k$-modes | 2779.42 ± 463.66 | 2.35 ± 0.07 | 0.15 ± 0.03 | 8 | 0 | 8545.5 | (649–15,532) |

Results for LCA and $k$-modes are reported as average ± standard deviation across five independent models.

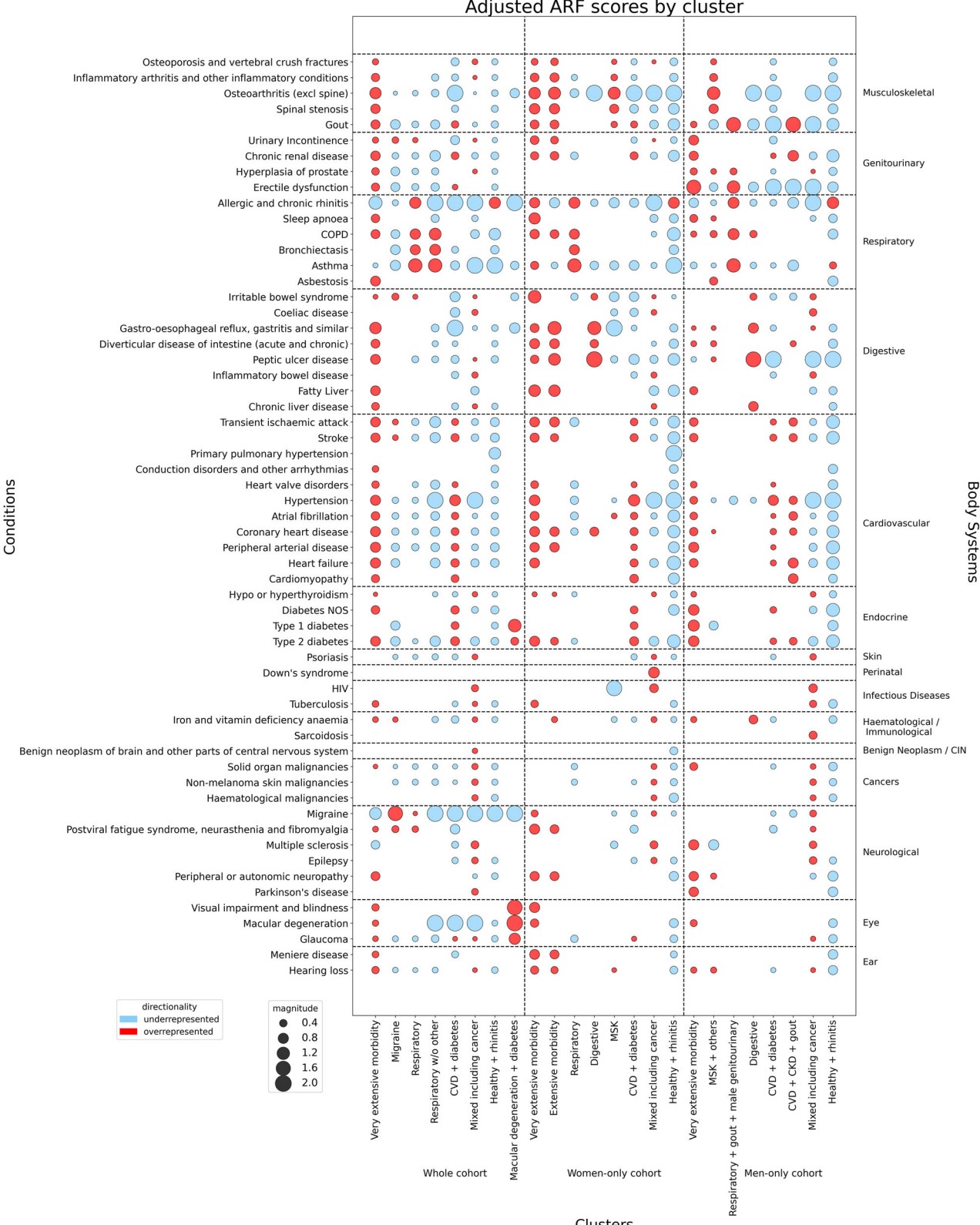

**Fig. 1 | Bubble heatmap shows under- and over-represented conditions in each cluster from the *k*-modes derived models.** Bubble sizes represent the magnitude of over- or under-representation. Magnitudes are computed as transformed adjusted relative frequency (ARF) values (Supplementary Note 2), which is an ordinal representation (*e.g.*, a red bubble of magnitude 2.0 indicates that a condition is more overrepresented than a red bubble of magnitude 1.5). Red bubbles indicate over-representation, while blue bubbles indicate under-representation. Only bubbles with a statistically significant over- or under-representation are shown, according to Fisher's Exact Test (*n* = 142,005 participants in the whole cohort, 77,785 participants in the women-only cohort 64,220 participants in the men-only cohort; *p* < 0.05).

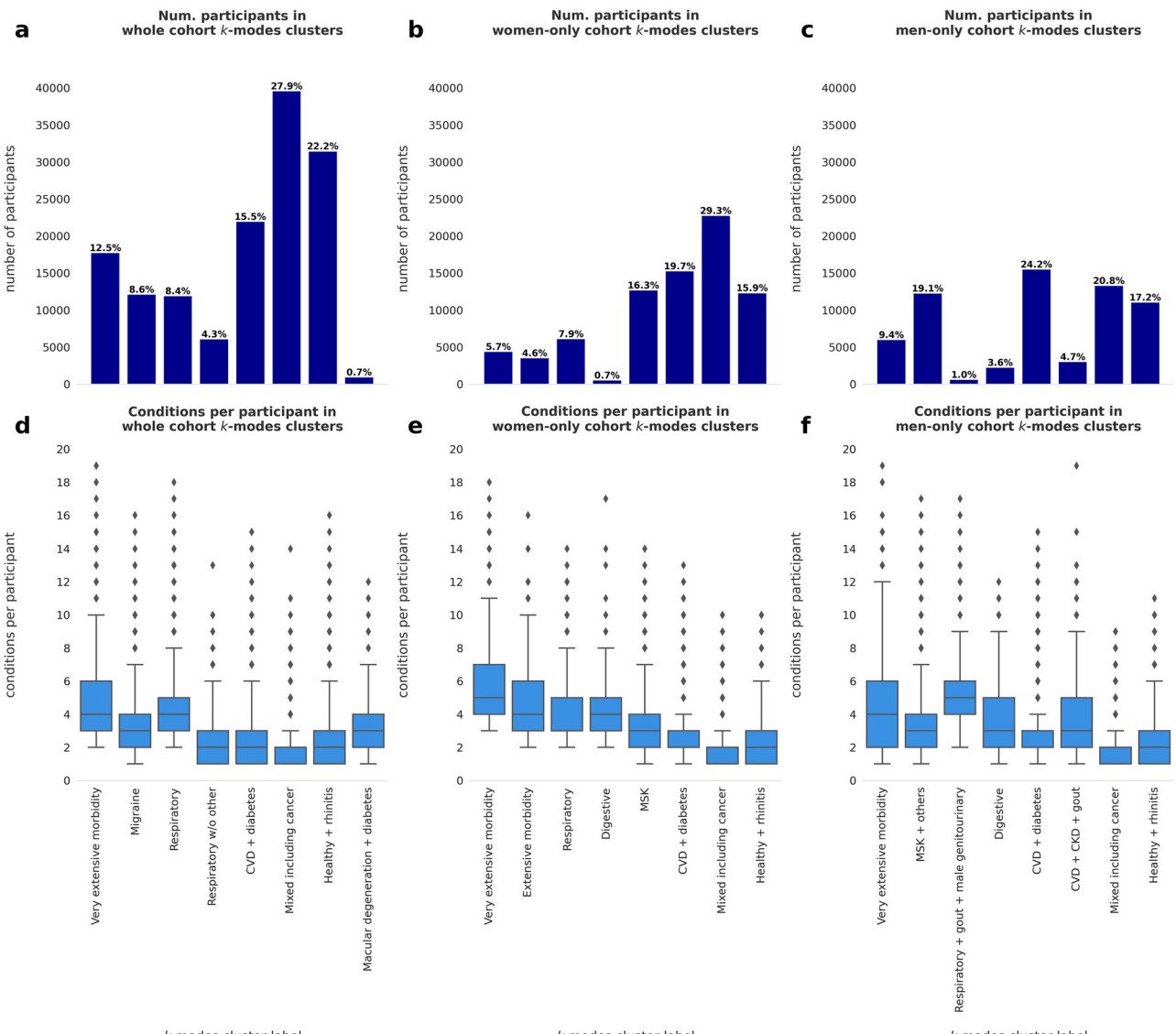

**Fig. 2 | Cluster sizes and condition counts. a** Sizes of clusters identified within the whole cohort. **b** Sizes of clusters identified within the women-only cohort. **c** Sizes of clusters identified within the men-only cohort. **d** Boxplots showing conditions per participant within each cluster of the whole cohort (*n* = 17,767; 12,142; 11,937; 6118; 21,989; 39,601; 31,477; 974 participants, respectively). **e** Boxplots showing conditions per participant within each cluster of the women-only cohort (*n* = 4420; 3549; 6133; 562; 12,715; 15,285; 22,787; 12,334 participants, respectively). **f** Boxplots showing conditions per participant within each cluster of the men-only cohort (*n* = 6023; 12,287; 649; 2281; 15,532; 3039; 13,341; 11,068 participants, respectively). Boxplot boxes show interquartile range (IQR), whereas boxplot whiskers, or error bars, show all data within 1.5x the IQR. Points above each boxplot represent outliers. Quartiles and IQR values per-cluster are available as Supplementary Data 3–5.

There were several consistencies across cohorts. The "*Very extensive morbidity*" clusters were the most strongly associated with depression in all three cohorts (whole: HR 2.42, 95% CI 2.17–2.69; women: HR 2.67, 95% CI 2.24–3.17; men: HR 2.65, 95% CI 2.22–3.18). Additionally, the "*Healthy + Rhinitis*" (whole: HR 1.59, 95% CI 1.46–1.75; women: HR 1.48, 95% CI 1.30–1.67; men: HR 1.50, 95% CI 1.29–1.75) and "*Mixed including cancer*" (whole: HR 1.62, 95% CI 1.48–1.77; women: HR 1.63, 95% CI 1.46–1.82; men: HR 1.60, 95% CI 1.38–1.86) clusters were generally the most weakly associated. Finally, association with depression also appeared to increase with the number of conditions per participant (Fig. 3). However, there were some exceptions; for example, the whole cohort's "*Macular degeneration + diabetes*" cluster had the fourth highest mean number of conditions (3.09), but was only weakly associated with depression (HR 1.29, 95% CI 0.85–1.98).

## Discussion
This study systematically explored clustering of physical health conditions using four methods appropriate for binary data (*k*-modes[33,34], *k*-medoids[24],

LCA[25] and AHC[23]). *K*-modes performed best, and the clusters identified were reasonably interpretable and often aligned with known associations between conditions. People with any physical condition at baseline were generally more likely to develop depression than people without any physical condition. There was some variation in this association by cluster, which may be at least partly driven by differences in the mean number of physical conditions in each cluster.

Existing studies of morbidity clustering typically apply a single method. One study compared LCA to a Bayesian, network-based approach, but used age and admission type, rather than conditions alone, to drive cluster formation[54]. Two other studies explored AHC and *k*-means in the same dataset, but chose *k*-means on the basis of AHC being too computationally intensive rather than based on performance[28,29]. Additionally, despite the use of *k*-means[55] by several multimorbidity studies[28–31], it typically relies upon Euclidean distance as its similarity measure[32], which is unsuitable for binary data[19]. Other multimorbidity studies have used *k*-means *after* a multiple correspondence analysis[56,57], which represents categorical features as a

**Table 2 | Hazard ratios per cluster for the development of subsequent depression**

| Cohort | Cluster Label | No. (%) of participants in cluster | Mean no. of conditions per participant in cluster | Hazard ratio (95% CI) |
|---|---|---|---|---|
| Whole cohort | Very extensive morbidity | 17,767 (12.5) | 4.50 | 2.42 (2.17, 2.69) |
| | Migraine | 12,142 (8.6) | 3.23 | 1.96 (1.75, 2.19) |
| | Respiratory | 11,937 (8.4) | 4.00 | 1.95 (1.74, 2.18) |
| | Respiratory w/o other | 6118 (4.3) | 2.29 | 1.86 (1.62, 2.15) |
| | CVD + Diabetes | 21,989 (15.5) | 2.38 | 1.78 (1.61, 1.97) |
| | Mixed including cancer | 39,601 (27.9) | 1.77 | 1.62 (1.48, 1.77) |
| | Healthy + Rhinitis | 31,477 (22.2) | 2.58 | 1.59 (1.46, 1.75) |
| | Macular degeneration + diabetes | 974 (0.7) | 3.09 | 1.29 (0.85, 1.98) |
| Women-only | Very extensive morbidity | 4420 (5.7) | 5.69 | 2.67 (2.24, 3.17) |
| | Extensive morbidity | 3549 (4.6) | 4.61 | 2.56 (2.12, 3.10) |
| | Respiratory | 6133 (7.9) | 3.72 | 1.93 (1.67, 2.23) |
| | Digestive | 562 (0.7) | 4.49 | 1.83 (1.14, 2.93) |
| | MSK | 12,715 (16.3) | 2.85 | 1.81 (1.58, 2.07) |
| | CVD + Diabetes | 15,285 (19.7) | 2.70 | 1.71 (1.52, 1.94) |
| | Mixed including cancer | 22,787 (29.3) | 1.75 | 1.63 (1.46, 1.82) |
| | Healthy + Rhinitis | 12,334 (15.9) | 2.02 | 1.48 (1.30, 1.67) |
| Men-only | Very extensive morbidity | 6023 (9.4) | 4.30 | 2.65 (2.22, 3.18) |
| | MSK + others | 12,287 (19.1) | 3.46 | 2.50 (2.15, 2.91) |
| | Respiratory + gout + male genitourinary | 649 (1.0) | 5.23 | 2.25 (1.43, 3.53) |
| | Digestive | 2281 (3.6) | 3.47 | 2.06 (1.60, 2.66) |
| | CVD + Diabetes | 15,532 (24.2) | 2.71 | 1.90 (1.65, 2.20) |
| | CVD + CKD + gout | 3039 (4.7) | 3.77 | 1.87 (1.47, 2.38) |
| | Mixed including cancer | 13,341 (20.8) | 1.62 | 1.60 (1.38, 1.86) |
| | Healthy + Rhinitis | 11,068 (17.2) | 2.08 | 1.50 (1.29, 1.75) |

All models were adjusted for baseline age, ethnicity, country of residence and deprivation.

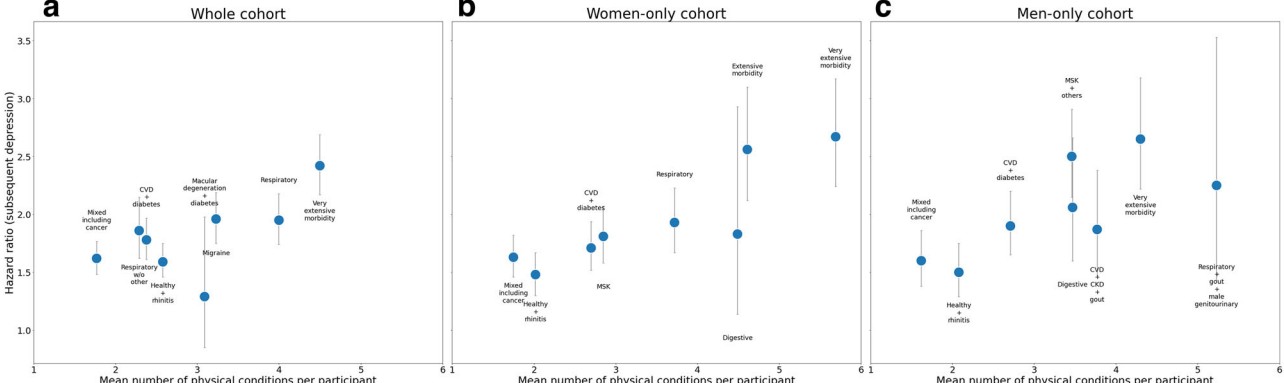

**Fig. 3 | Risk of subsequent depression by mean number of physical conditions.** **a** Hazard ratio versus mean number of physical conditions for the whole cohort. **b** Hazard ratio versus mean number of physical conditions for the women-only cohort. **c** Hazard ratio versus mean number of physical conditions for the men-only cohort. Each point represents a cluster value. Error bars represent the 95% confidence interval of the hazard ratio.

low-dimensional Euclidean space[30,31]. While this transforms the data features into an appropriate format for $k$-means, it also manipulates the data based on their pairwise co-occurrences, which may not be appropriate for every dataset.

This study finds that almost all physical morbidity clusters are associated with higher risk of subsequent depression than the group with no physical conditions at baseline. Although the strength of association varied by cluster, this seemed to be partly explained by the mean number of conditions in the cluster. This is consistent with a similar study[30], which

found associations between severe mental illness and a higher number of physical conditions. Another similar study, which aimed to identify groups of physical conditions associated with incident depression within a Taiwanese cohort, also found that social factors played a role on the risk of subsequent depression diagnosis[22]. Specifically, they found that amongst four "Cardiometabolic", "Arthritis-cataract", "Multimorbidity" and "Relatively healthy" clusters, those within the "Arthritis-cataract" and "Multimorbidity" clusters had significantly higher risk of depression than healthy individuals. However, this association was attenuated for participants who

engaged in social activities, including a job, volunteer experience or community activities[22].

Strengths of this study include the analysis of a large dataset which records morbidities in both baseline research data and linked routine data, as well as the inclusion of a wide set of morbidities recommended by a recent consensus study[36]. Notably, this study is unique for its implementation and comparison between four clustering methods appropriate for binary data. A limitation is that the data are collected from volunteers who are generally more affluent than the UK average, and people from ethnic minorities are somewhat under-represented[58]. Additionally, our study does not consider genitourinary conditions, which are exclusive to women. However, given the middle-aged demographic of participants, several such conditions pertaining to pre-menopausal women are less applicable[59] while those pertaining to peri- and post-menopausal women are poorly coded[60]. Furthermore, there is no standard way to evaluate the validity of identified clusters, although the observed clusters do include several known clinical associations. Due to the limited time for which we were granted access to the UKB, we could not compare cluster patterns obtained from other clustering methods. Consequently, this warrants further validation studies in other datasets to explore reproducibility of cluster solutions. Finally, we could not account for depression severity during follow-up because such information is not well recorded in the available health records; future research should investigate the relationship between physical multimorbidity and subsequent depression severity.

Many previous studies of morbidity clustering do not provide much information about which conditions are over or under-represented in clusters, which leaves readers relying solely on author-chosen cluster labels for interpretation[20,22]. For example, it is common for other studies to identify a 'cardiometabolic' cluster[22] and the "*CVD + diabetes*" cluster in our study was amongst the three largest clusters in all three cohorts. However, it is not straightforward to compare clusters across studies because of considerable variation in the conditions included in analysis, and because many clustering studies do not provide detailed information about the nature of identified clusters. Key implications are that clustering studies should be more consistent in the choice of conditions to include (and, at a minimum, follow consensus recommendations[36]). Additionally, they should report the nature of clusters to help understand them beyond their high-level labels (by, for example, visualising the prevalence of individual conditions in each cluster alongside over/under representation). We believe that our ARF measure with visualisation in a bubble heatmap demonstrates one way to do this. However, there is a need for multimorbidity researchers to develop improved and consistent cluster visualisations and explanations to facilitate interpretation and to enhance clinical utility.

Morbidity clustering studies also typically use one clustering method, but there is no single clustering method which is likely to be optimal for every dataset. We therefore believe that clustering studies should more systematically explore different methods and make explicit how they choose the best method for their datasets and purposes. To encourage similar systematic comparison of different cluster methods, we have provided access to our code (https://github.com/laurendelong21/clusterMed)[49]. Many studies also cluster the entire population, which is likely not sensible given the very different incidence and prevalence of disease with age and, to a lesser extent, sex and ethnicity. In this analysis, study participants were mostly middle-aged (so we did not further stratify by age) and overwhelmingly white, but we found some differences in the clusters identified in the whole population versus women or men separately. Although whole population clustering may be appropriate in some circumstances, reporting of clusters stratified by age and sex (and ethnicity if the data permits) would be valuable to explore how clustering varies by demographic characteristics.

Finally, further research to better understand why physical multimorbidity is associated with subsequent depression is needed. The general trend between increased risk of subsequent depression and mean number of conditions suggests a social explanation: suffering more conditions may more strongly interrupt one's life or sense of self. However, the relationship between depression and physical conditions is very likely bidirectional and longitudinal research, which better examines how the two interact over a lifetime, particularly from a cohort with younger participants, would be valuable[61].

## Conclusions

Using the best performing of four different clustering methods, this study identified several multimorbidity clusters which align with known clinical associations. Association with depression varied between clusters, but this may be partly driven by differences in the number of conditions. More research is needed to better understand the mechanisms underlying such associations.

## Data availability

The UKB data is not openly available to protect the privacy of participants. Researchers can register for access here: https://www.ukbiobank.ac.uk/enable-your-research/register. The numerical values underlying main text Fig. 1, including adjusted *p*-values, are included in Supplementary Data 2. The numerical values underlying Fig. 2d–f, are included in Supplementary Data 3–5. All other data underlying Figs. 2, 3 are included in Tables 1, 2.

## Code availability

Corresponding code is available at https://github.com/laurendelong21/clusterMed[49].

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

## Acknowledgements
This work was co-funded by the Medical Research Council and the National Institute for Health Research (grant number MR/S028013/1), and the NIHR AIM-CISC programme (grant number NIHR202639). The views expressed are those of the author(s) and not necessarily those of the NIHR or the Department of Health and Social Care. The study was conducted using the UKB Resource under application number 57213. LND is individually funded by a Global Informatics Scholarship from the School of Informatics at the University of Edinburgh. The School of Informatics had no role in study design, data collection and analysis, decision to publish, or preparation of the manuscript. The authors would like to thank the UKB participants and the UKB staff for their contributions to this study. The authors would like to thank the public members of our advisory board, Dr Paul Kelly and Pat Watson, for providing thoughtful feedback throughout our project. This work has made use of the resources provided by the Edinburgh Compute and Data Facility (ECDF) (http://www.ecdf.ed.ac.uk/).

## Author contributions
All authors contributed to writing the manuscript. L.N.D. and K.F. conducted the analyses and made the figures. L.N.D. and P.G. wrote the software. K.F. and R.P. processed and prepared the data. J.D.F. and B.G. designed and supervised the study.

## Competing interests
The authors declare no competing interests.
