## [Transparent Peer Review file · Communications Medicine]

Cluster and survival analysis of UK Biobank data reveals associations between physical multimorbidity clusters and subsequent depression

Corresponding Author: Ms Lauren DeLong

Version 0:

Reviewer comments:

Reviewer #1

(Remarks to the Author)

Thank you for this interesting manuscript. The authors have completed a tremendous amount of work in assessing multimorbidity patterns using different techniques.

The authors completed three performance metrics to assess the suitability and performance of four different clustering techniques and have chosen K-modes to further examine the hazard ratio for depression over time. It is unsure if the performance metrics reflect model fit and diagnostic criteria. It is desirable to see the model fit and diagnostic criteria for the model with eight clusters from the k-modes technique, which seems to be selected based on Calinski and Harabasz, and Davies-Bouldin scores for the whole population. LCA has the highest score when applying the same criteria to the sex-specific cohort.

It would also be useful to report on the cluster patterns for (one) other clustering methods (e.g. LCA) and how they predict subsequent depression.

Below are several specific comments which will enhance the manuscript.

Introduction: Line 48: Please include the prevalence of depression.

Models and metrics: The rationale for measuring the magnitude of over- and under-represented conditions within a cluster relative to the whole cohort prevalence is unclear. Please clarify.

Figure 2: cluster size, please include the % of each cluster.

Overall, this is an insightful manuscript. Thank you.

Reviewer #2

(Remarks to the Author)

This is an important study considering the rising incidence of depression. The investigation revealed that certain clusters of physical multimorbidity, particularly cardiometabolic conditions, were associated with a higher risk of depression, highlighting the need for further research into social factors that may contribute to these links. I suggest authors categorize depression into mild, moderate, and severe, and examine whether the associations between clusters and different severities of depression vary.

Reviewer #3

(Remarks to the Author)

This research paper reports investigation of different statistical methods to identify multimorbidity clusters in the UK Biobank dataset in order to predict subsequent occurrence of depression. The topic is interesting and timely as multimorbidity is increasing worldwide and depression is one of the main conditions contributing to this tendency (e.g. <https://doi.org/10.1016/j.eclinm.2023.101860>). Besides the strength of the paper, namely to evaluate different statistical methods, the major weakness is the selection of diseases to investigate that should be addressed.

1. From the data point of view:

- a. The selected conditions are not balanced in the depth of ICD diagnosis. For example, there are conditions that correspond to diagnostic categories well (e.g. sleep apnoea, migraine, psoriasis) but others are collections of several diseases (e.g. solid organ malignancies).
- b. Female specific genitourinary diseases are lacking.
- c. Although the authors state that they followed "previously published lists" their selection deviates from these at several points. However, it was not explained why some conditions were omitted.
- d. In addition, specific features of the UK Biobank dataset were not taken into account. For example, disease categories were included in the analyses that contained very low number (<1%) of participants, e.g. Down's syndrome (n=48 in the whole UKB), Thalassaemia (n=7660), myasthenia gravis (n=595). Numbers must be even lower in the selected sample.

2. From depression point of view:

- a. Long-term physical health conditions were selected not considering the previous literature of comorbidities of depression. Several studies investigated the likely comorbidities of depression (e.g. <https://doi.org/10.4088/JCP.14m09147>, <https://doi.org/10.1371/journal.pcbi.1005487>), even analysing possible genetic overlaps (e.g. <https://doi.org/10.1016/j.xgen.2023.100371>, <https://doi.org/10.1186/s13073-021-00927-6>).
- b. Predicting subsequent occurrence of depression in a population aged 37-73 years may mean that clusters where depression onset is early the HR of depression will be lower than in real clinical observations. Thus it is possible that mean number of physical conditions per participants would be less important in a younger cohort. Therefore investigating the longitudinal aspects of comorbidity would be important, as the author also stated (e.g. <https://doi.org/10.1038/s41467-024-51467-7>).

Minor point: Readers without bioinformatics expertise might benefit from an explanation of the advantages and disadvantages of the compared methods. In the supplementary materials it was explained with mathematical terms but to introduce it to e.g. medical doctors or neuroscientists would increase the number of interested readers.

Version 1:

Reviewer comments:

Reviewer #2

(Remarks to the Author)

Reviewer #3

(Remarks to the Author)

The authors fully addressed my points, and I have no further questions or suggestions.

Revisions for Manuscript COMMSMED-24-0679

To the reviewers and editors:

Thank you very much for taking the time to read and provide feedback on our manuscript, "Investigating associations between physical multimorbidity clusters and subsequent depression: cluster and survival analysis of UK Biobank data". We appreciate your positive comments as well as your thoughtful suggestions. Below, we describe the steps we took to address each of your remarks and recommendations. We hope you will all find that we have addressed each piece of feedback thoroughly. Thank you for your time and the opportunity to revise our manuscript.

Sincerely, Ms. Lauren Nicole DeLong and co-authors.

Reviewer #1 (Remarks to the Author):

1. Thank you for this interesting manuscript. The authors have completed a tremendous amount of work in assessing multimorbidity patterns using different techniques.

Authors' Response:

Thank you for dedicating time to reading and understanding our paper thoroughly.

2. The authors completed three performance metrics to assess the suitability and performance of four different clustering techniques and have chosen K-modes to further examine the hazard ratio for depression over time. It is unsure if the performance metrics reflect model fit and diagnostic criteria.

Authors' Response:

The performance metrics, Calinski and Harabsz score, Davies-Bouldin score, and Silhouette score, all describe cluster quality within a model. In summary, they explain the extent to which similar data points cluster together and the degree of separation between dissimilar data points. These metrics are described, in detail, within Supplementary Appendix 1. However, to make this clearer in the main text, we amend lines 106-107 as follows:

"To assess suitability and performance of these clustering methods, we used three performance metrics (Calinski and Harabsz score ⁴², Davies-Bouldin score ⁴³, and Silhouette score ⁴⁴ (Supplementary Appendix 1)), which are appropriate and commonly used for assessing cluster quality in an unsupervised setting."

In contrast, measuring *model fit*, as the reviewer mentioned, is more appropriate for *supervised* machine learning rather than *unsupervised* machine learning. The four clustering approaches we used

are unsupervised. In an unsupervised setting, there are no established or known labels for the data. In other words, the true underlying cluster patterns are not known *a priori*. Through unsupervised learning methods like the four we explored, we can investigate possible clusters underlying the data, and different methods give different results. The three performance metrics are used, therefore, to assess inter- and intra- cluster quality, rather than fit to known labels.

3. It is desirable to see the model fit and diagnostic criteria for the model with eight clusters from the *k*-modes technique, which seems to be selected based on Calinski and Harabasz, and Davies-Bouldin scores for the whole population. LCA has the highest score when applying the same criteria to the sex-specific cohort.

Authors' Response:

The *k*-modes approach was selected based upon the best Calinski and Harabasz, Davies-Bouldin, and Silhouette scores. As clarified in point 2, these scores measure cluster quality, and they are reported within Table 1 of our manuscript.

Regarding the reviewer's request for model fit: as mentioned in point 2, model fit is more appropriate for supervised machine learning. In contrast, the four clustering approaches we explored are unsupervised techniques in which there are no established or known labels for the data. The three aforementioned scores are used, therefore, to assess inter- and intra- cluster quality, rather than fit to known labels.

Although it may appear that LCA has the highest Calinski and Harabasz scores within the sex-specific cohorts, this difference is not pronounced when accounting for the variation, which is reported in Table 1 as standard deviations.

4. It would also be useful to report on the cluster patterns for (one) other clustering methods (e.g. LCA) and how they predict subsequent depression.

Authors' Response:

Thank you for the suggestion. Unfortunately, the duration of time for which we were granted access to the UK Biobank data has since expired, so we no longer have patient-level data.

However, with our publicly available code (<https://github.com/laurendelong21/clusterMed>), one may reproduce the results of this work and investigate other clustering methods.

5. Introduction: Line 48: Please include the prevalence of depression.

Authors' Response:

We have now included the prevalence of depression on line 48:

“Depression affects an estimated 6% of people worldwide...[11]”

[11] Malhi, G. S. & Mann, J. J. Depression. The Lancet 392, 2299–2312 (2018).

6. Models and metrics: The rationale for measuring the magnitude of over- and under-represented conditions within a cluster relative to the whole cohort prevalence is unclear. Please clarify.

Authors' Response:

Thank you for raising this point. We have now added the following sentence on pages 6-7, lines 112-114:

“By adjusting for cohort prevalence, the ARF accounts for conditions which are *generally* rare or common, facilitating more effective comparisons between clusters than prevalence alone (Supplementary Fig. 2).”

7. Figure 2: cluster size, please include the % of each cluster.

Authors' Response:

We have now revised this figure to include the percentage of participants within each cluster as an annotation above each respective bar.

8. Overall, this is an insightful manuscript. Thank you.

Authors' Response:

Thank you for your feedback.

Reviewer #2 (Remarks to the Author):

9. This is an important study considering the rising incidence of depression. The investigation revealed that certain clusters of physical multimorbidity, particularly cardiometabolic conditions, were associated with a higher risk of depression, highlighting the need for further research into social factors that may contribute to these links.

Authors' Response:

Thank you for taking the time to read our manuscript. We are glad that the contributions are clear.

10. I suggest authors categorize depression into mild, moderate, and severe, and examine whether the associations between clusters and different severities of depression vary.

Authors' Response:

Thank you for the suggestion. In the United Kingdom, depression is usually diagnosed in the primary care setting and thus most of the depression diagnoses during follow-up come from the primary care records. Unfortunately, primary care records do not tend to differentiate between mild, moderate and severe depression, and so we cannot categorize depression in this way. We agree that future research should examine the relationship between physical multimorbidity and depression, and so we have added the following sentence on page 17, lines 258-260:

“Finally, we could not account for depression severity during follow-up because such information is not well recorded in the available health records; future research should investigate the relationship between physical multimorbidity and subsequent depression severity.”

Reviewer #3 (Remarks to the Author):

11. This research paper reports investigation of different statistical methods to identify multimorbidity clusters in the UK Biobank dataset in order to predict subsequent occurrence of depression. The topic is interesting and timely as multimorbidity is increasing worldwide and depression is one of the main conditions contributing to this tendency (e.g. <https://doi.org/10.1016/j.eclinm.2023.101860>).

Authors' Response:

Thank you for reading our manuscript and for recognizing its importance. We have also now referenced the relevant study you mentioned:

3. Chowdhury, S. R., Chandra Das, D., Sunna, T. C., Beyene, J. & Hossain, A. Global and regional prevalence of multimorbidity in the adult population in community settings: a systematic review and meta-analysis. *EClinicalMedicine* 57, 101860 (2023).

12. Besides the strength of the paper, namely to evaluate different statistical methods, the major weakness is the selection of diseases to investigate that should be addressed.

Authors' Response:

We hope that our responses to the next points sufficiently address the reviewer's remarks.

1. From the data point of view:

13. a. The selected conditions are not balanced in the depth of ICD diagnosis. For example, there are conditions that correspond to diagnostic categories well (e.g. sleep apnoea, migraine, psoriasis) but others are collections of several diseases (e.g. solid organ malignancies).

Authors' Response:

We agree with the reviewer that some conditions are treated at a more granular level than others. As described in more detail in Prigge *et al.* [35], three clinicians involved in this project selected long-term conditions of relevance to middle-aged adults (the recruited population in UK Biobank) from a previously published list of 308 conditions. After reaching clinical consensus, 154 conditions were selected which were all at the same level of granularity.

Importantly, the research presented in our manuscript is part of a larger project co-funded by the Medical Research Council and the National Institute for Health Research through grant number

MC/S028013. As part of this larger project, we were interested in the bi-directional relationship between depression and physical multimorbidity. As a result, some conditions had to be grouped to avoid double-counting of the same underlying condition (for example, ‘unstable angina’, ‘stable angina’, ‘myocardial infarction’, and ‘coronary heart disease unspecified’ were all included in a single condition - ‘coronary heart disease’) in longitudinal analyses that were planned as part of the funded project.

In summary, these groupings occur to avoid double-counting of closely related conditions (*e.g.* different types of cancer) or conditions that evolve over time (*e.g.* juvenile arthritis in adolescents may be recorded as rheumatoid arthritis once they are adults). These details are clarified within Prigge *et al.* [35].

35. Prigge, R. et al. Robustly Measuring Multiple Long-Term Health Conditions Using Disparate Linked Datasets in UK Biobank. Preprints with The Lancet (2024). Under review.

<http://dx.doi.org/10.2139/ssrn.4863974>

14. b. Female specific genitourinary diseases are lacking.

Authors’ Response:

The reviewer raises a valid point. We would like to note, though, that cervical, ovarian and uterine cancers are grouped within the “solid organ malignancies” condition. We have also now included the following sentences on page 17, lines 252-255, to acknowledge your point:

“Additionally, our study does not consider genitourinary conditions which are exclusive to women. However, given the middle-aged demographic of participants, several such conditions pertaining to pre-menopausal women are less applicable while those pertaining to peri- and post-menopausal women are poorly coded.”

15. c. Although the authors state that they followed “previously published lists” their selection deviates from these at several points. However, it was not explained why some conditions were omitted.

Authors’ Response:

As mentioned within point 13, some conditions were grouped in Prigge *et al.* [35] to avoid double-counting of closely related conditions. In addition, a Delphi consensus study [34] recommended several conditions which should *always* or *usually* be included within multimorbidity measurement. From the ‘always include’ list, our study includes all but one condition: ‘Paralysis (other than stroke)’ was excluded because our decision on which conditions to include preceded the consensus study. Conditions on the ‘usually include’ list were included in the project unless they were not applicable to our specific study

goals (e.g., acute conditions or conditions not applicable to the age range of participants). Again, these details are clarified within Prigge *et al.* [35].

In addition, there may be several other reasons why there appear to be deviations:

- Firstly, as mentioned in lines 79-81, we only use physical conditions in our study: “We ascertained the presence of depression and 69 long-term physical health conditions at baseline using data from the baseline visit and from linked GP, hospital, and cancer registry records based on previously published lists.” As part of our study design, this helps us to investigate the associations between physical multimorbidity and depression.
- Additionally, as explained within the next point of this rebuttal (point 16), conditions were omitted from visualizations if they were not informative (determined by ARF and Fisher’s Exact Test) for distinguishing between clusters. However, despite these conditions being omitted from visualizations, they were included in the clustering analyses.

We hope that these explanations, in addition to our response to point 13, clarify upon your remark.

34. Ho, I. S. S. et al. Measuring multimorbidity in research: Delphi consensus study. *BMJ Medicine* 1, e000247 (2022).

35. Prigge, R. et al. Robustly Measuring Multiple Long-Term Health Conditions Using Disparate Linked Datasets in UK Biobank. Preprints with *The Lancet* (2024). Under review.
<http://dx.doi.org/10.2139/ssrn.4863974>

16. d. In addition, specific features of the UK Biobank dataset were not taken into account. For example, disease categories were included in the analyses that contained very low number (<1%) of participants, e.g. Down’s syndrome (n=48 in the whole UKB), Thalassemia (n=7660), myasthenia gravis (n=595). Numbers must be even lower in the selected sample.

Authors’ Response:

We appreciate the reviewer’s remark regarding the inclusion of rarer conditions in the analyses. We believe that our methods sufficiently account for cluster differences amongst rarer conditions.

As clarified within point 6 of this response, we analyze cluster differences using adjusted relative frequency (ARF), which reports the cluster-specific prevalence of a condition relative to its prevalence in the whole cohort. Thus, with ARF, a condition which is generally rare across the population can be descriptive for distinguishing between cluster condition profiles.

Within the same “Models and Metrics” section (page 6), we explain that we also use Fisher’s Exact test to assess whether such differences are statistically significant. As explained in Supplementary Appendix 2, conditions which have no significant under- or over- representation are omitted from the visualization

(Fig. 2). As seen in Fig. 2, Down's syndrome, for example, helps distinguish one cluster (women-only, *Mixed including cancer*) from others. On the other hand, Thalassemia and myasthenia gravis were omitted from the visualization, indicating that they were uninformative for distinguishing between clusters. Therefore, our methodology (ARF + Fisher's Exact Test) inherently accounts for features like condition prevalence.

We hope that our clarification in response to point 6, as well as our responses on condition choice within points 13-15 make this clearer.

2. From depression point of view:

17. a. Long-term physical health conditions were selected not considering the previous literature of comorbidities of depression. Several studies investigated the likely comorbidities of depression (e.g. <https://doi.org/10.4088/JCP.14m09147>, <https://doi.org/10.1371/journal.pcbi.1005487>), even analysing possible genetic overlaps (e.g. <https://doi.org/10.1016/j.xgen.2023.100371>, <https://doi.org/10.1186/s13073-021-00927-6>).

Authors' Response:

Thanks for pointing out these relevant studies to us. We hope that our responses to points 13 to 15 further explain our rationale for why/how we selected our conditions. We would also like to highlight that we did include many of the conditions mentioned within the studies suggested above. Below, we describe the overlap between these studies and ours:

- The first study (<https://doi.org/10.4088/JCP.14m09147>) lists pain, constipation, multiple sclerosis, viral hepatitis, Parkinson's disease, and migraine as the most comorbid with depression.
 - Our study included multiple sclerosis, Parkinson's disease, and migraine. These conditions can be found within Supplementary Table 1.
 - Our study also included chronic viral hepatitis as a sub-condition of chronic liver disease.
 - Constipation can be a symptom of many of the diseases of the digestive system, and pain can be a symptom of many of the included conditions, in general. Therefore, constipation and pain are, implicitly, included.
- The second study (<https://doi.org/10.1371/journal.pcbi.1005487>) found that anxiety related disorders (anxiety, panic attack, stress, nervous breakdown), and postnatal depression are highly comorbid with depression.
 - Due to our study goal to investigate the associations between physical conditions and depression, we excluded anxiety related disorders as these tend to be mental health disorders, rather than physical disorders.

- Additionally, since our study population included participants which were mostly above child-bearing age (ages 37-73), postnatal depression is also less applicable to our study.
- While the third study (<https://doi.org/10.1016/j.xgen.2023.100371>) investigates multimorbidity in the UK Biobank dataset, it is not entirely clear to us which conditions are specifically comorbid to depression.
- The fourth study (<https://doi.org/10.1186/s13073-021-00927-6>) provides a browser to explore the most-associated comorbidities (<https://mai.fudan.edu.cn/multimorbidity/browse/info?id=111&name=F33%20Recurrent%20depressive%20disorder>). "Recurrent depressive disorder," the closest category they include to depression, is most associated with:
 - "Specific personality disorders" and "anxiety disorders" which we excluded as they are not physical conditions.
 - "Other arthritis": although this is a general category, we believe we have covered this, since we include different types of arthritis within our category, "Musculoskeletal conditions" (Supplementary Table 1).
 - "Other functional intestinal disorders": again, this is a rather general category which we believe to be covered by our category, "Diseases of the Digestive System" (Supplementary Table 1).
 - "Chronic renal failure", which we include ("Chronic renal disease" within "Diseases of the Genitourinary system" (Supplementary Table 1)).

Furthermore, the selection of physical conditions, independent of previous literature, was an intentional decision, given our study design. Clustering approaches are unsupervised, so there are no pre-defined classifications imposed upon the data. Instead, unsupervised clustering approaches are designed to find patterns or sub-groups underlying the data based upon the selected features. Therefore, such methods are ideal for discovering patterns which may be previously unknown.

If we were to select conditions that are suggested by previous works as comorbid with depression, we would impose external biases that may eliminate the possibility of discovering new patterns of comorbidity through the clustering approaches. Additionally, by being more inclusive with the selection of conditions, our study may serve as independent validation for any previous studies with which our results align.

To make this clearer, we have included the following sentence on page 6, lines 101-102:

“All four clustering methods are designed to operate without pre-defined classifications, making them ideal for discovering underlying groups within the data.”

18. b. Predicting subsequent occurrence of depression in a population aged 37-73 years may mean that clusters where depression onset is early the HR of depression will be lower than in real clinical observations. Thus it is possible that mean number of physical conditions per participants would be less important in a younger cohort. Therefore investigating the longitudinal aspects of comorbidity would be important, as the author also stated (e.g. <https://doi.org/10.1038/s41467-024-51467-7>).

Authors' Response:

We agree that longitudinal aspects are an important prospective direction, particularly to investigate whether our observed results are reproducible within a younger cohort. Thus, we have revised the corresponding sentence in our discussion (page 19, line 293) to:

“However, the relationship between depression and physical conditions is very likely bidirectional, and longitudinal research which better examines how the two interact over a lifetime, particularly from a cohort with younger participants, would be valuable [61].”

We also added the referenced paper as a citation:

61. Gezsi, A. et al. Unique genetic and risk-factor profiles in clusters of major depressive disorder-related multimorbidity trajectories. *Nat Commun* 15, 7190 (2024).

19. Minor point: Readers without bioinformatics expertise might benefit from an explanation of the advantages and disadvantages of the compared methods. In the supplementary materials it was explained with mathematical terms but to introduce it to e.g. medical doctors or neuroscientists would increase the number of interested readers.

Authors' Response:

Thank you for this helpful point. We have now included the following text in Supplementary Appendix 1:

“Each of these four methods has different properties. For instance, k-modes assigns groups based on similarity between the participants of a cluster, whereas k-medoids assigns groups based on similarity to a single, representative participant of each cluster. LCA is unique because it gives the probabilities of each participant belonging to each cluster. These three methods, though, require prior specification of cluster number, which is typically unknown. In contrast, AHC does not require any pre-specified cluster number; instead, cluster number is determined empirically. However, AHC is limited in that it assumes the data can be separated hierarchically.”

Final Revisions for Manuscript COMMSMED-24-0679A

To the reviewers and editors:

Thank you very much for taking the time to read and provide feedback on our manuscript, which is now titled: "Cluster and survival analysis of UK Biobank data reveals associations between physical multimorbidity clusters and subsequent depression ". Below, we describe the most recent revisions made. Thank you for your time and the opportunity to revise our manuscript.

Sincerely, Ms. Lauren Nicole DeLong and co-authors.

Reviewer #3 (Remarks to the Author):

1. The authors fully addressed my points, and I have no further questions or suggestions.

Authors' Response:

Thank you for reading our paper and providing helpful feedback.

Editor (Remarks to the Author):

2. We therefore invite you to revise your paper one last time to explain that you could not do the additional analysis requested by Reviewer 1 in the last round as a limitation.

Authors' Response:

Thank you for the opportunity to revise again. To address this limitation, we now include the following lines on page 15, lines 258-260 of our manuscript:

“Due to the limited time for which we were granted access to the UK Biobank, we could not compare cluster patterns obtained from other clustering methods. Consequently, this warrants further validation studies in other datasets to explore reproducibility of cluster solutions.”

3. At the same time we ask that you edit your manuscript to comply with our format requirements and to maximise the accessibility and therefore the impact of your work.

Authors' Response:

We have now altered our manuscript and related materials to comply with the specifications set out in the Editorial Requests Table. We hope you will find that we have fulfilled all editorial requests. If you find anything which needs to be addressed further, please let us know, and we are happy to discuss again.